# Impact of Signet-Ring Cell Histology in the Management of Patients with Non-Metastatic Gastric Cancer: Results from a Retrospective Multicenter Analysis Comparing FLOT Perioperative Chemotherapy vs. Surgery Followed by Adjuvant Chemotherapy

**DOI:** 10.3390/cancers15133342

**Published:** 2023-06-25

**Authors:** Riccardo Giampieri, Maria Giuditta Baleani, Alessandro Bittoni, Francesca Rastelli, Vincenzo Catalano, Michela Del Prete, Silvia Chiorrini, Giada Pinterpe, Francesco Graziano, Francesca Chiara Giorgi, Renato Bisonni, Rosarita Silva, Paolo Alessandroni, Lara Mencarini, Rossana Berardi

**Affiliations:** 1Medical Oncology Unit, Dipartimento Scienze Cliniche e Molecolari, Università Politecnica delle Marche and Azienda Ospedaliero-Universitaria Ospedali Riuniti delle Marche, 60126 Ancona, Italyr.berardi@staff.univpm.it (R.B.); 2Department of Oncology, Ospedale Generale Provinciale, 62100 Macerata, Italy; maria.baleani@sanita.marche.it; 3Department of Medical Oncology, IRCCS Istituto Romagnolo per lo Studio dei Tumori (IRST) “Dino Amadori”, 47014 Meldola, Italy; 4Department of Oncology, Ospedale “C.e G. Mazzoni” Ascoli Piceno, 63100 Ascoli Piceno, Italy; 5Department of Oncology, Ospedale Santa Maria della Misericordia, AV1, 61029 Urbino, Italy; 6Department of Oncology, Ospedale Augusto Murri di Fermo, 63900 Fermo, Italy; 7Department of Oncology, Ospedale E. Profili, 60044 Fabriano, Italy; 8Department of Oncology, Azienda Ospedaliera Marche Nord, AV1, 61121 Pesaro, Italy; 9Department of Oncology, Ospedale Madonna del Soccorso, 63074 San Benedetto del Tronto, Italy

**Keywords:** gastric cancer, FLOT, signet ring, adjuvant chemotherapy, neoadjuvant chemotherapy

## Abstract

**Simple Summary:**

FLOT-based perioperative chemotherapy is the mainstay of treatment for patients with non-metastatic gastric cancer in Western countries. Although signet-ring cell histology has been proven to be associated with a lack of response to chemotherapy, there is a lack of data concerning whether FLOT-based chemotherapy might be less effective in patients with signet-ring histology. The aim of this retrospective study is to assess whether patients whose gastric cancer is signet-ring cell positive might benefit less compared to others when treated with perioperative FLOT.

**Abstract:**

Background: FLOT perioperative chemotherapy represents the standard of care in non-metastatic gastric cancer patients. Signet-ring cell positivity is associated with a worse prognosis in patients with gastric cancer treated with chemotherapy. Comparison between FLOT perioperative chemotherapy vs. surgery followed by adjuvant chemotherapy based on signet-ring cell positivity is lacking. The aim of the analysis was to compare perioperative FLOT with adjuvant chemotherapy in gastric cancer patients stratified by signet-ring cell positivity. Methods: We conducted a retrospective multicenter analysis based on disease-free survival (DFS) and overall survival (OS) in patients with gastric cancer who received perioperative chemotherapy with a FLOT regimen and compared their survival with a historical cohort of patients treated with adjuvant chemotherapy, matched by cT and cN stage and by tumor histological features. Results: Seventy-six patients were enrolled and 24 (32%) were signet-ring cell positive. At a median follow-up time of 39 months, the median DFS was 26.3 months and the median OS was 37.3 months. Signet-ring cell positivity was associated with a shorter OS (median OS: 20.4 vs. 46.9 months, HR: 3.30, 95%CI: 1.56–6.99, *p* = 0.0018) and DFS (mDFS: 15.2 vs. 38.6 months, HR: 3.18, 95%CI: 1.55–6.54, *p* = 0.0016). This was confirmed by multivariate analysis for DFS (Exp(B): 2.55) and OS (Exp(B): 2.68). After propensity score matching, statistically significant shorter DFS (HR: 3.30, 95%CI: 1.50–7.35, *p* = 0.003) and OS (HR: 5.25, 95%CI: 2.18–12–68, *p* = 0.0002) were observed for patients with signet-ring cell positivity who received perioperative treatment vs. those who received surgery followed by adjuvant chemotherapy. Conclusions: Signet-ring positivity was associated with shorter DFS and OS in patients who received perioperative treatment with FLOT compared with surgery followed by adjuvant therapy. These data suggest that for patients with signet-ring cell histology, FLOT perioperative treatment might not always be the best choice of treatment, and further research should be focused on this group of patients.

## 1. Introduction

Gastric cancer incidence has decreased considerably in the past decade, yet it is still one of the deadliest gastrointestinal tumors [1]. Owing to the lack of proper screening programs in Western countries, most patients are diagnosed in an advanced disease stage.

Indeed, early gastric cancer diagnoses are relatively uncommon, and most patients with non-metastatic gastric cancer require careful multidisciplinary management [2]; even though surgery still represents the mainstay of treatment and the only chance for cure, pre and/or post-operative chemotherapy is also advised, to reduce the high relapse risk.

Eastern countries’ guidelines suggest that surgery followed by adjuvant chemotherapy should be used: earlier diagnoses, more favorable tumor biology and high surgical volumes for this disease support this kind of approach. In Western countries, it is suggested that perioperative chemotherapy should be used [3].

After the FLOT4 trial [4], FLOT rapidly became the standard perioperative treatment for this setting. The addition of Ramucirumab, such as in the RAMSES FLOT7 trial [5], did not yield any additional gain in activity, and molecularly selected treatment approaches are currently under investigation [6,7] but do not represent the standard of care.

With these considerations, it is crucial to identify those patients who do not benefit from perioperative treatment with FLOT: these patients are those on which research should be focused, to develop tailored treatment choices in those instances where perioperative treatment with FLOT seems to be less active.

Several different clinical factors, such as extent of disease involvement and different tumor histology, have been associated with differences in survival outcomes. Previous published data have suggested that diffuse-type gastric cancer and the presence of signet-ring cells in tumors might be associated with a lack of response to standard chemotherapy [8,9,10]. Signet-ring cell histology has also been described as a poor prognostic factor, particularly in patients with an advanced stage of disease, whereas its role as a prognostic factor in patients diagnosed with early-stage disease is not so well defined [11,12,13,14].

The aim of our study was to assess the impact of different histological features in patients with non-metastatic gastric/GEJ cancer who received perioperative FLOT. We compared the outcomes of patients treated with perioperative FLOT, stratified by histology, with an historical cohort of patients with gastric cancer who received surgery followed by adjuvant therapy. This comparison was performed to assess whether tumor histological features might have a different effect on survival outcomes for patients who received either treatment strategy.

## 2. Materials and Methods

### 2.1. Patients

We conducted a retrospective analysis on patients who received perioperative treatment with a FLOT regimen for non-metastatic gastric/GEJ cancer. This multicenter analysis enrolled patients treated in seven different oncology departments in hospitals located in the Marche Region (Clinica Oncologica–Ospedali Riuniti delle Marche, Oncologia–Ospedale Murri di Fermo, Oncologia–Azienda Ospedaliera Marche Nord, Oncologia–AV1, Oncologia–Ospedale di Fabriano Profili and Oncologia–Ospedale di Ascoli Piceno and Oncologia–Ospedale di San Benedetto). Patients were treated from 2017 until September 2022 (time of data cut-off).

Patients’ demographics were collected based on sex (male vs. female), age and clinical T and N stage at treatment start. Only patients with cT3 or higher T stage and/or cN+ stage were subsequently analyzed.

Performance status by ECOG, radicality of surgery, adjuvant therapy omission (ATOM) and pathological complete response in the tumor specimen (yes vs. no) were also recorded.

Histological features that were used as stratifying factors were based on Lauren classification (intestinal-type gastric cancer/IGC vs. diffuse-type gastric cancer/DGC). In particular, we defined intestinal-type/diffuse-type gastric cancer as per Lauren classification. Mixed-type gastric cancers were excluded from calculations based on histology because of not being able to consider them as either intestinal or diffuse-type gastric cancer. Signet-ring cell positive tumors were those where the presence of signet-ring cells was observed in more than 50% of a tumor sample.

Patients in the historical cohort of patients who received surgical resection followed by adjuvant therapy were enrolled based on the availability of data concerning stage of diagnosis and on the basis of having received adjuvant therapy for either pT3 or higher and/or pN-positive gastric cancer. Patients in the historical cohort were collected from 2005 until 2020.

### 2.2. Methods

The aim of this analysis was to assess differences in DFS among patients with non-metastatic gastric cancer who received perioperative treatment with a FLOT regimen, stratified by histology.

Disease-free survival (DFS) was calculated by the Kaplan–Meier method, starting from the beginning of neoadjuvant therapy until the first clinical sign of relapse/death for those patients who were not lost at follow-up. As for patients who progressed into metastatic/unresectable disease at the end of neoadjuvant treatment or those who were found to have metastases/unresectable disease upon surgery, the time of first discovery of metastatic disease/unresectable disease was used as an end-point for further calculations. We also calculated overall survival (OS) by the Kaplan–Meier method, starting from the beginning of neoadjuvant therapy until death for those patients who were not lost at follow-up. DFS for patients in the historical cohort was calculated by the Kaplan–Meier method, starting from the date of surgery until the first clinical sign of relapse/death for those patients who were not lost at follow-up. OS for patients in the historical cohort was calculated by the Kaplan–Meier method, starting from the date of surgery until death for those patients who were not lost at follow-up.

Survival differences among stratification factors were assessed by a log-rank test. Multivariate analysis was conducted by Cox-proportional model hazard regression.

Since we aimed at comparing survival outcomes with a historical group of patients who received surgery followed by adjuvant therapy vs. those patients who received surgery after neoadjuvant therapy with FLOT, we conducted a matching process in this cohort of patients to reduce the weight of confounding factors. Propensity score matching was used [15], with the “Nearest” method and ratio 1: 1. The matching variables were cT and cN stage, diffuse-type histology and signet-ring cell presence/absence.

For all analyses, the level of statistical significance *p* was set at 0.05.

All tests were conducted using MedCalc^®^ Statistical Software version 19.7.2 (MedCalc Software Ltd., Ostend, Belgium; https://www.medcalc.org (accessed on 8 May 2023); 2021) and R software for Windows (4.1.0 Beta Build, packages used: survival, survminer and MatchIt).

The study was submitted and approved by the local ethical committee (Comitato Etico Regione Marche) on 19 May 2022, protocol no. 2022 144.

## 3. Results

### 3.1. Patients’ Characteristics

Seventy-six patients were enrolled in this study.

In terms of stage at the start of perioperative treatment, 38/76 (50%) had cT4 stage, whereas 59/76 (77%) had cN+ stage. Most patients (57/76, 75%) had ECOG PS 0 at the start of perioperative treatment.

Some 53/76 (70%) of patients were male, and the remaining 23/76 (30%) were female. The median age of the cohort was 64 (range 38–78). Only 4/76 (5%) of patients were older than 75 years old.

In terms of histology, 24/76 (32%) had a diffuse-type histology by Lauren classification, 23/76 (30%) had an intestinal-type histology and 4/76 (5%) had a mixed histology. Some 24/76 (32%) of patients also had signet-ring cells described within their tumor sample, whereas 25/76 (33%) patients had indetermined gastric cancer. Of the indetermined gastric cancer, undifferentiated (G3) gastric cancer (by WHO classification) was the most frequent subtype (22/76, 29%).

In 21/76 (27%) of patients the tumor was located in the cardia/gastroesophageal junction (GEJ), 5 (6%) patients had a tumor located in the gastric fundus, and the remaining 50/76 (67%) had tumors located in the gastric body or antro-pyloric region.

Some 70/76 (92%) of patients were able to complete all scheduled four cycles of FLOT neoadjuvant treatment. Two patients only received one cycle of FLOT neoadjuvant treatment and had to stop neoadjuvant treatment and undergo surgery due to gastric bleeding. One patient suffered from acute myocardial infarction after the first cycle of treatment with FLOT and after that he discontinued neoadjuvant treatment. Two patients received only three cycles of neoadjuvant treatment and went on to receive surgery due to worsening clinical conditions. One patient suffered from febrile neutropenia after the first cycle of FLOT neoadjuvant treatment and continued for the remaining three cycles of treatment with FOLFOX instead.

Only 66/76 (85%) of patients were ultimately resected after preoperative treatment. Among these 65 patients, 54 received R0 resection, 9 received R1 resection and 3 received R2 resection. Some 6/66 (10%) patients achieved complete pathological remission in their tumor specimen.

Out of 66 patients who received surgery after perioperative chemotherapy, only 57 (86%) received adjuvant chemotherapy after preoperative management. Three patients were lost at follow-up after surgery and did not receive any post-operative treatment. Five patients had such poor clinical conditions due to weight loss and loss of muscle mass after surgery that they were deemed not able to receive any post-operative treatment. One patient died after surgery due to reasons not directly associated to the disease or treatment (traumatic injury).

A summary of patients’ characteristics can be found in Table 1.

### 3.2. Univariate Survival Analysis

The median follow-up time of this group of patients was 39 months. During this follow-up time, 42/76 (55%) patients had relapsed or progressed directly into metastatic disease after perioperative treatment, and 40/76 (52%) had already died.

Median disease-free survival (DFS) of the whole cohort of patients was 26.3 months, whereas median overall survival (OS) was 37.3 months.

Positive cN stage was not associated with differences in DFS (mDFS: 22.5 months vs. NR, HR: 1.68, 95%CI: 0.82–3.44, *p* = 0.16). Positive cN stage was also not associated with differences in OS (mOS: 33.8 vs. NR, HR: 1.76, 95%CI: 0.83–3.73, *p* = 0.14).

cT4 stage was associated with shorter DFS (mDFS: 18.6 months vs. NR, HR: 2.38, 95%CI: 1.24–4.54, *p* = 0.0087) (Figure 1a). cT4 stage was also associated with shorter OS (mOS: 22.7 months vs. NR, HR: 2.86, 95%CI: 1.48–5.52, *p* = 0.0017) (Figure 1b).

Diffuse histology was not associated with shorter DFS (mDFS: 18.1 vs. 29.3 months, HR: 1.27, 95%CI: 0.65–2.47, *p* = 0.475). Diffuse histology was also not associated with shorter OS (mOS: 24.3 months vs. 38.6 months, HR: 1.49, 95%CI: 0.74–3.01, *p* = 0.26).

Undifferentiated histology was not associated with shorter DFS (mDFS: 20.2 vs. 38.6 months, HR: 1.86, 95%CI: 0.96–3.63, *p* = 0.06). Undifferentiated histology was associated instead with shorter OS (mOS: 23.8 vs. 41.0 months, HR: 2.19, 95%CI: 1.08–4.45, *p* = 0.0299).

Signet-ring cell positivity was associated with shorter DFS (mDFS: 15.2 vs. 38.6 months, HR: 3.31, 95%CI: 1.60–6.83, *p* = 0.0012) (Figure 2a). Signet-ring cell positivity was also associated with shorter OS (mOS: 20.4 vs. 46.9 months, HR: 3.30, 95%CI: 1.56–7.00, *p* = 0.0018) (Figure 2b).

Lack of adjuvant therapy (ATOM) was not associated with shorter DFS (mDFS: 9.3 vs. 37.3 months, HR: 2.30, 95%CI: 0.75–7.10, *p* = 0.1448). ATOM was also not associated with shorter OS (mOS: 17.1 vs. 46.9 months, HR: 2.74, 95%CI: 0.85–8.85, *p* = 0.092).

Performance status as assessed by the Eastern Cooperative Oncology Group (ECOG PS) (0 vs. 1–2) was not associated with differences in DFS (mDFS: 29.3 vs. 20.2 months, HR: 0.72, 95%CI: 0.36–1.46, *p* = 0.37). ECOG PS (0 vs. 1–2) was also not associated with differences in OS (mOS: 41.0 vs. 22.6 months, HR: 0.62, 95%CI: 0.29–1.33, *p* = 0.22).

Radicality of surgery (R0 vs. R1 vs. R2) was associated with differences in DFS (mDFS: 51.9 vs. 11.8 vs. 1.2 months, respectively, *p* < 0.0001). Radicality of surgery (R0 vs. R1 vs. R2) was also associated with differences in OS (mOS: 46.9 vs. 24.2 vs. 5.0 months, respectively, *p* < 0.0001).

Complete response after neoadjuvant therapy was not associated with statistically significant differences in DFS (mDFS NR vs. 33.8 months, HR: 0.39, 95%CI: 0.14–1.10, *p* = 0.0762). Complete response after neoadjuvant therapy was also not associated with statistically significant differences in OS (mOS NR vs. 41.0 months, HR: 0.41, 95%CI: 0.13–1.29, *p* = 0.128).

Tumor location (GEJ vs. gastric cancer) was not associated with differences in DFS (mDFS: 26.3 vs. 23.8 months, HR: 0.90, 95%CI: 0.44–1.84, *p* = 0.79). Tumor location was also not associated with differences in OS (mOS: 38.6 vs. 37.3 months, HR: 0.82, 95%CI: 0.41–1.64, *p* = 0.58).

### 3.3. Multivariate Survival Analysis and Comparison to Adjuvant Treatment

Signet-ring cell histology (Exp(B): 3.21, 95%CI: 1.37–7.56, *p* = 0.0072) maintained its statistically significant impact on DFS at multivariate analysis. Other variables that were associated at univariate analysis with differences in DFS were not confirmed at multivariate analysis, with the exception of surgical radicality.

Signet-ring cell histology (Exp(B): 2.59, 95%CI: 1.09–6.15, *p* = 0.0309) and cT4 stage (Exp(B): 2.33, 95%CI: 1.03–5.25, *p* = 0.0409) maintained their statistically significant impact on OS at multivariate analysis. Other variables that were associated at univariate analysis with differences in OS were not confirmed at multivariate analysis, with the exception of surgical radicality.

A summary of both univariate and multivariate analysis can be found in Table 2.

Patients’ characteristics from the historical cohort of patients that was used for comparison can be found in the Appendix A. In the historical cohort of patients, 1 patient received R2 resection, whereas out of the remaining 94 patients, 76 (81%) received R0 resection and 16 (17%) patients received R1 resection. There was a statistically significant impact on OS (mOS, respectively, 76.0 vs. 53.7 vs. 11.8 months for R0 vs. R1 vs. R2, *p* < 0.0001)

Some 95 patients from the historical cohort were ultimately matched. A histogram (Appendix A) and jitter plot (Appendix A) of the results of matching procedure can be found in the Appendix A.

After the matching procedure, when assessing differences in survival among patients with signet-ring cell histology, patients who received perioperative FLOT treatment had a shorter overall survival compared with those patients who received surgery followed by adjuvant therapy (mOS: 23.8 vs. 73.4 months, HR: 5.14, 95%CI: 1.80–14.67, *p* = 0.0022) (Figure 3b). DFS was also significantly shorter (mDFS: 18.6 vs. 26.7 months, HR: 2.79, 95%CI: 1.12–6.95, *p* = 0.0274) (Figure 3a).

On the other hand, OS in patients who did not have signet-ring cell histology was not significantly different comparing patients who received FLOT perioperatively to those who received surgery followed by adjuvant chemotherapy (mOS: 57.0 vs. 59.8 months, HR: 2.15, 95%CI: 0.94–4.88, *p* = 0.0674). DFS was also not significantly different between these two groups of patients (mDFS: 51.9 vs. 51.7 months, HR: 1.71, 95%CI: 0.83–3.54, *p* = 0.144).

## 4. Discussion

Gastric cancer treatment guidelines suggest that multidisciplinary management is crucial in patients with non-metastatic disease involvement [16]: accurate staging is required to avoid unnecessary surgery, and several tumor-related issues such as dysphagia and relevant weight loss must be accounted for. After radical surgery, a high proportion of patients will experience disease relapse and will ultimately receive chemotherapy with palliative intent. Indeed, only a few selected patients will also receive locoregional therapy aiming to prolong survival [17,18,19].

In order to reduce the risk of disease relapse, perioperative or post-operative treatment, usually by means of chemotherapy, is strongly advised. Radiotherapy might also be considered, particularly in those cases where locoregional disease extent seems to be particularly severe.

Since the MAGIC trial [20], most guidelines in Western countries suggest that a perioperative approach should be preferred to surgical resection followed by adjuvant therapy. It must be said that there was no definitive consensus regarding whether perioperative treatment should be used in all patients [21,22]. These doubts have mostly disappeared after the results of the FLOT4 trial [4] had been published, thus leading to considering FLOT perioperative treatment as the standard of care for this group of patients.

Even though the FLOT4 trial has demonstrated the superiority of perioperative FLOT vs. ECF/ECX chemotherapy, a few issues still remain; indeed, the impact of signet-ring cell positivity as a prognostic factor in gastric cancer patients might not be so well-defined, as it has been suggested that it might be a marker of better prognosis in early-stage gastric cancer, opposed to having a poor prognostic value in patients with more advanced stages of disease involvement [23,24]. This might be explained due to the fact that signet-ring cell positivity might increase tumor resistance to chemotherapy. On this basis, it could be hypothesized that signet-ring cell positivity might also decrease perioperative FLOT chemotherapy effectiveness.

The aim of our analysis was to assess whether FLOT perioperative treatment might have worse results compared to surgery followed by adjuvant therapy in patients whose tumors had signet-ring cell positivity, due to a lack of effect on neoadjuvant treatment.

Based on the centers that participated in this analysis, it can be safely assumed that our data fully summarize everyday clinical practice in the whole Marche region during the years 2017–2021, for this setting.

Interestingly, cN stage had only a minor impact on survival outcomes, even though a trend towards improved survival was seen in patients with cN0 stage upon perioperative treatment start. We believe that the relatively small number of patients with cN0 status might have contributed to not being able to observe a statistically significant impact on OS, although we cannot rule out that other factors might have also contributed to that. In addition to that, achievement of complete pathological response was not significantly associated with differences in survival outcomes, although a trend towards better overall survival was observed. These results are different from other published studies that have reported the prognostic role of tumor regression grade (TRG) for gastric cancer patients treated with neoadjuvant chemotherapy [25,26]. It should be noticed that all these studies have used rather heterogeneous chemotherapy options different from FLOT; thus a comparison with the results of our study are limited. In addition to that, the number of patients who achieved complete pathological regression in our study cohort is rather limited (6/76, 8%), and that might have reduced the prognostic impact of TRG on our study cohort.

As expected, the achievement of surgical radicality was one of the factors that maintained its role as a predictor of different survival: albeit neoadjuvant therapy is able to determine tumor shrinkage to some extent, and cure is only achieved by means of radical surgery; for those patients who are still not optimally resected after significant tumor shrinkage, survival outcomes are similar to those of patients treated with palliative intent for metastatic disease.

This factor might explain why cT4 stage before beginning neoadjuvant treatment was such a relevant prognostic factor in our cohort of patients: despite some patients potentially experiencing tumor size reduction, whenever the locoregional extent of the disease seems to be not just limited to the gastric wall (as in the case of cT4 stage), the chance of achieving radical surgical resection remains quite small. The quality of surgical assessment before treatment start is therefore crucial, as it defines the likelihood of radicality: in our group of patients who were initially deemed able to undergo radical surgery, only 66/76 (85%) were ultimately resected and only 51/66 (77%) achieved R0 resection. The difference in likelihood to receive R0/R1/R2 resection was not significantly different between the group of patients who received FLOT neoadjuvant treatment followed by surgery followed by FLOT adjuvant treatment vs. patients who received surgery followed by adjuvant treatment (*p* = 0.3078). On the other hand, if we consider the whole population of patients, to also include patients who received neoadjuvant FLOT and who did not receive any surgery, there was a statistically significant difference in overall survival compared with patients who received surgery followed by adjuvant treatment (*p* = 0.0017). We believe that this difference might have been related to sub-optimal staging before the start of neoadjuvant treatment, as all patients received a chest-abdomen CT scan and echoendoscopy but only a few patients also received a laparoscopy to exclude the presence of peritoneal metastases at the start. We also believe that part of this difference in survival might have been related to too much optimistic assessment of the likelihood to respond to FLOT neoadjuvant treatment that might have led surgeons to deem as resectable tumors that would prove to be ultimately not resectable.

Finally, the most relevant finding of our analysis is that signet-ring cell gastric cancer patients might derive less benefit from neoadjuvant therapy with FLOT: indeed, patients with signet-ring cell histology had poorer survival if they received perioperative treatment compared with those who underwent surgical resection and post-operative chemotherapy.

These data suggest a cautious approach when suggesting perioperative treatment with FLOT in patients whose tumors harbor features that might reduce the likelihood of response to chemotherapy. Other studies have suggested similar results: patients whose tumors harbor microsatellite instability/defective mismatch repair proteins expression where the benefit of 5FU-based chemotherapy might be reduced achieve best results when they undergo surgical resection rather than receiving preoperative treatment [27].

We acknowledge the limitations of our analysis: matching procedures should reduce potential confounding factors, but there might be some confounding factors that cannot be accounted for even with matching analysis. In particular, the rate of surgery of the two cohorts of patients was different (only 85% of the perioperative treatment group vs. 100% of the historical cohort), and this factor might partly explain differences in survival.

Despite these limitations, the suggestion that signet-ring cell and diffuse-type cancers might derive less benefit from neoadjuvant therapy with FLOT should not be simply dismissed; there are indeed a few papers that suggest that perioperative treatment might still be less effective in signet-ring cell histology.

Messager et al. [28] conducted a retrospective cohort study of perioperative chemotherapy in patients treated from 1997 to 2010 focused on signet-ring cell histology. The authors found that despite the fact that a higher percentage of patients in the perioperative treatment had received adjuvant therapy compared to those who received up-front surgery (66.5% vs. 35.2%, *p* < 0.0001), a shorter survival was observed in patients who received perioperative chemotherapy (12.8 vs. 14.0 months, *p* = 0.043). This was maintained at multivariate analysis as a predictor of worse overall survival (HR: 1.4, 95%CI: 1.1–1.9, *p* = 0.0.42).

Similar to our results, a recently published paper [29] compared patients with signet-ring cell carcinoma treated with neoadjuvant therapy vs. those treated with surgery followed by adjuvant therapy. After propensity score matching, despite having used chemotherapy regimens different from the FLOT regimen (SOX, DOS, Xelox, Folfox6 and DOX), patients with signet-ring cell cancer who received neoadjuvant treatment had worse survival outcomes compared with those patients who received up-front surgery (5 -year OS rates for neoadjuvant chemotherapy vs. surgery were 50% and 64.7%, respectively, *p* = 0.192).

To conclude, we believe that our data suggest that patients with non-metastatic gastric cancer with signet-ring cell histology and diffuse-type cancer should be assessed more thoroughly, as they have higher chance of strategy failure when receiving perioperative treatment; this fact should be considered as novel treatment strategies focused on total neoadjuvant therapy, performed by means of prolonged neoadjuvant therapy, have recently been suggested [30]. As our data suggest that patients with signet-ring cell histology might benefit less from neoadjuvant chemotherapy, it might also be expected that total neoadjuvant therapy might have even greater limitations in this subset of patients, as recently published data seem to suggest [31]. Other treatment strategies, perhaps focused on molecular pathways that are associated with signet-ring cell histology such as those targeting CDH1 pathway, are needed for this group of patients.

## 5. Conclusions

Perioperative management of patients with non-metastatic gastric cancer has improved considerably in the last decade. The only limitation of this kind of approach is the lack of proper personalization of treatment based on different tumor biology. Recently, novel treatment options have been tested in this setting, namely anti-HER2 drugs for HER2-positive patients and immune checkpoint inhibitors mainly for patients with MSI-H tumors. Based on the results of our study, we believe that patients with signet-ring-positive cells should be investigated in detail, as the results of standard perioperative treatment with FLOT seem to be particularly lacking in this setting. Investigation into molecular targets that are usually involved in signet-ring cell gastric cancer, such as CDH1, is strongly advised.

## Figures and Tables

**Figure 1 cancers-15-03342-f001:**
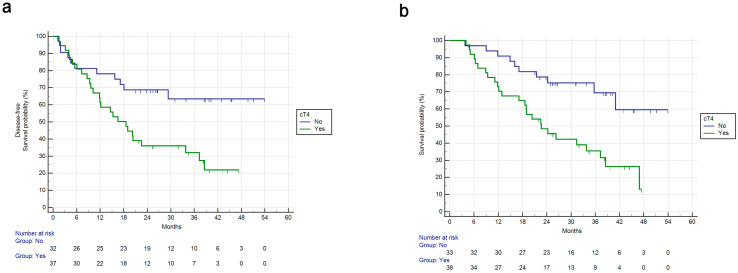
Disease-free survival and overall survival for patients in the FLOT perioperative cohort based on cT4 stage upon treatment start or not. (**a**): DFS for perioperative FLOT in cT4 stage vs. not. mDFS: 18.6 months vs. NR, HR: 2.38, 95%CI: 1.24–4.54, *p* = 0.0087. (**b**): OS for perioperative FLOT in cT4 stage vs. not. mOS: 22.7 months vs. NR, HR: 2.86, 95%CI: 1.48–5.52, *p* = 0.0017.

**Figure 2 cancers-15-03342-f002:**
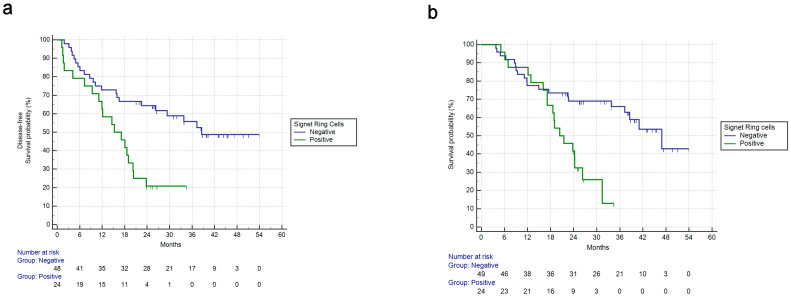
Disease-free survival and overall survival for patients in the FLOT perioperative cohort based on signet-ring cell positivity or not. (**a**): DFS for perioperative FLOT in signet-ring cell positivity vs. not. mDFS: 15.2 vs. 38.6 months, HR: 3.31, 95%CI: 1.60–6.83, *p* = 0.0012. (**b**): OS for perioperative FLOT in signet-ring cell positivity vs. not. mOS: 20.3 vs. 46.9 months, HR: 3.30, 95%CI: 1.56–7.00, *p* = 0.0018.

**Figure 3 cancers-15-03342-f003:**
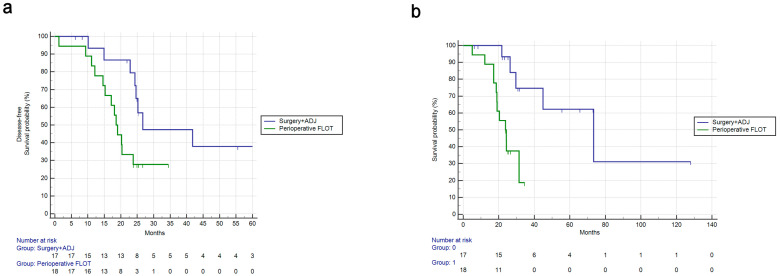
Disease-free survival and overall survival for patients with signet-ring positivity in the FLOT perioperative cohort vs. adjuvant chemotherapy cohort. (**a**): Results for DFS comparison. mDFS: 18.6 vs. 26.7 months, HR: 2.79, 95%CI: 1.12–6.95, *p* = 0.0274. (**b**): Results for OS comparison. mOS: 23.8 vs. 73.4 months, HR: 5.14, 95%CI: 1.80–14.67, *p* = 0.0022.

**Table 1 cancers-15-03342-t001:** Summary of patients’ clinical characteristics.

Characteristic	N (%) Tot = 76
Sex	
male	53 (70%)
female	23 (30%)
Age	
≥75	4 (5%)
45–75	64 (85%)
≤45	8 (10%)
Primary tumor location	
GEJ	21 (28%)
fundus	5 (6%)
body	44 (57%)
pylorus	6 (9%)
cT before treatment start	
cTx	7 (9%)
cT1	1 (1%)
cT2	4 (5%)
cT3	26 (34%)
cT4a	35 (46%)
cT4b	3 (4%)
cN before treatment start	
cNx	2 (3%)
cN0	15 (20%)
cN+	59 (77%)
ypT after surgery	
ypT0	6 (8%)
ypTis	1 (1%)
ypT1a/b	6 (8%)
ypT2	3 (4%)
ypT3	28 (37%)
ypT4a	17 (22%)
ypT4b	5 (6%)
not resected	10 (13%)
ypN after surgery	
ypN0	20 (26%)
ypN1	11 (14%)
ypN2	15 (20%)
ypN3	20 (26%)
not resected	10 (13%)
Surgical radicality	
R0	54 (71%)
R1	9 (12%)
R2	3 (4%)
not resected	10 (13%)
Adjuvant therapy started	
yes	57 (75%)
no	9 (12%)
not resected	10 (13%)
ECOG PS upon neoadjuvant start	
0	57 (75%)
1	15 (20%)
2	3 (4%)
unknown	1 (1%)
Histotype by Lauren	
intestinal-type	23 (30%)
diffuse-type	24 (32%)
mixed	4 (5%)
indetermined-type	25 (33%)
Signet-ring cells	
yes	24 (32%)
no	52 (68%)

**Table 2 cancers-15-03342-t002:** Results of univariate and multivariate analyses (in bold where statistically significant).

Univariate							Multivariate					
Factor	OS(HR)	95%CI	p	DFS(HR)	95%CI	p	OS(HR)	95%CI	p	DFS(HR)	95%CI	p
**cN+ vs. cN0**	1.76	0.83–3.73	0.14	1.68	0.82–3.44	0.16	/	/	/	/	/	/
**cT4 vs. not**	**2.86**	**1.48–5.52**	**0.0017**	**2.38**	**1.24–4.54**	**0.0087**	**2.33**	**1.03–5.25**	**0.0409**			
**Diffuse vs. not**	1.49	0.74–3.01	0.26	1.27	0.65–2.47	0.475	/	/	/	/	/	/
**G3 vs. not**	**2.19**	**1.08–4.45**	**0.029**	1.86	0.96–3.63	0.06	1.07	0.48–2.38	0.86	/	/	/
**Signet ring vs. not**	**3.30**	**1.56–7.00**	**0.0018**	**3.31**	**1.6–6.83**	**0.0012**	**2.59**	**1.09–6.15**	**0.309**	**3.21**	**1.37–7.56**	**0.0072**
**ATOM vs. not**	2.74	0.85–8.85	0.092	2.30	0.75–7.10	0.1448	/	/	/	/	/	/
**ECOG PS 0 vs. 12**	0.62	0.29–1.33	0.22	0.72	0.36–1.46	0.37	/	/	/	/	/	/
**Radicality (R0 vs. R1 vs. R2)**	**/**	**/**	**<0.0001**	**/**	**/**	**<0.0001**	**24.01** **(R0–R1 vs. R2)**	**3.72–154.92**	**0.0008**	**62.64** **(R0–R1 vs. R2)**	**8.78–447.07**	**<0.0001**
**CR vs. not**	0.41	0.13–1.29	0.128	0.39	0.14–1.10	0.0762	/	/	/	/	/	/
**GEJ vs. other**	0.82	0.41–1.64	0.58	0.90	0.44–1.84	0.79	/	/	/	/	/	/

List of abbreviations: G3 = undifferentiated, ATOM = adjuvant therapy omission, ECOG PS = Eastern Cooperative Oncology Group performance status, CR = complete pathological response, GEJ = gastroesophageal junction.

## Data Availability

Data used for this study are freely available to be shared by the corresponding author upon reasonable request.

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
