# Peer review of "Impact of Signet-Ring Cell Histology in the Management of Patients with Non-Metastatic Gastric Cancer: Results from a Retrospective Multicenter Analysis Comparing FLOT Perioperative Chemotherapy vs. Surgery Followed by Adjuvant Chemotherapy"

_cancers, 2023, doi:10.3390/cancers15133342_

Round 1

Reviewer 1 Report

I would like to thank the editor for the opportunity to review this interesting paper comparing perioeperative chemotherapy vs. adjuvant chemotherapy for the treatment of signet-cell gastric and GEJ cancer.

Below are my comments:

1- In the methods section the authors write: "Propensity score matching was used, with “Nearest” method and ratio 1:1. Matching variables were cT and cN stage,  diffuse-type histology and signet-ring cells presence/absence. 

Can the authors specify the caliper that they used for matching?

I would also suggest the authors to add "age" as a matching variable. There are initial evidences about patients age >80 that after starting FLOT therapy can achieve a surgical resection only in 25% of cases.

2- Can the authors specify if all patients treated with FLOT achieved the full preoperative 4 cycles. If not, can they give some details about reduction or interruption of FLOT therapy?

3- In the results the authors state:"Out of 66 patients who received surgery after perioperative chemotherapy, only 57 160 (86%) received adjuvant chemotherapy after pre-operative management. "

Can the authors add some details about the reasons why patients did not received adjuvant FLOT?

4- Please provide a table about univariate and multivariate analysis results.It will be easier for the readers instead of going through all the text.

5- In the discussion section the authors pointed out that: " Interestingly, cN stage had only a minor impact on survival outcomes." How can the authors explain this contradictory result comparing it with the available evidences?

6- Concerning the surgical radically the authors comment in the discussion : "Quality of surgical assessment before treatment start is therefore crucial, as it defines the likelihood of radicality: in our  group of patients that were initially deemed able to undergo radical surgery, only 66/76 302 (85%) were ultimately resected, thus suggesting that there is dire need of improvement in this aspect. "

How can the authors explain a surgical radically of 71% with a ypT4a of 22% and a ypT4b of only 6%? Did they experience a R1 or R2 resection of all T4 patients?

This data could be a tremendous bias about survival analysis. Moreover I could not find, even in the supplementary materials, any data about radically od surgery in the historical cohort of patients. Which was the R0 resection rate in the adjuvant CHT group?

7- Please pay attention about the Privacy Policy of this journal. In the Supplementary 2 Adjuvant cohort database there are sensitive data (name of the patients, date of birth, etc...)

Minor polishing needed.

Author Response

We thank you for your constructive criticism. Please find below the replies to your comments:

1- In the methods section the authors write: "Propensity score matching was used, with “Nearest” method and ratio 1:1. Matching variables were cT and cN stage,  diffuse-type histology and signet-ring cells presence/absence. 

Can the authors specify the caliper that they used for matching?

REPLY: We decided not to use a caliper, as the relatively small number of patients that were enrolled (76) would have been reduced even more by using caliper and we hypothised that this would reduce the likelihood of being able to assess properly differences in outcome between the different groups of patients due to lack of appropriate sample size. However, following your suggestion, we tested caliper=0.2 on our analyses. Propensity score matching adjusted by such caliper was able to match only 55 patients in each group to be compared. In the newly defined subset of patients treated with FLOT, signet-ring cell positivity maintained a statistically significant impact on OS (mOS respectively 23.77 vs 46.92 months, HR:4.75, 95%CI:1.73-13.04, p=0.0025), DFS (mDFS respectively 15.24 vs 51.93 months, HR:3.98, 95%CI:1.56-10.11, p=0.0037)  

I would also suggest the authors to add "age" as a matching variable. There are initial evidences about patients age >80 that after starting FLOT therapy can achieve a surgical resection only in 25% of cases.

REPLY: Thank you for your suggestion. In the group of patients treated with FLOT, we did not find patients aged more than 80 years old. We used as cut-off for matching 75 years old threshold instead. Performing propensity score matching (caliper=0.2) and matching variables age >=75 y.o., cT4 yes vs not, cN+ vs not, diffuse-type histology yes vs not and signet-ring cells absence vs not, 55 patients in FLOT cohort and historical cohort were matched. In this newly defined subset of patients, signet-ring cell histology was still associated with worse OS when patients received perioperative management with FLOT opposed to surgery followed by adjuvant chemotherapy (mOS respectively 23.77 vs 73.37 months, HR:4.25, 95%CI:1-50-12.02, p=0.0064). In the group of patients who received perioperative management with FLOT and who were older than 75 y.o., mOS was not significantly different compared to patients aged less than 75 y.o. (mOS respectively NR vs 38.55 months, p=0.154). It must be said that all patients that were older than 75 y.o. and that were matched were able to receive surgery, differently from data that you have cited.

2- Can the authors specify if all patients treated with FLOT achieved the full preoperative 4 cycles. If not, can they give some details about reduction or interruption of FLOT therapy?

REPLY: Thank you for asking this important question: out of 76 patients that started FLOT perioperative treatment, 6 (8%) patients were not able to complete all 4 cycles of neoadjuvant FLOT. 2 patients only received one cycle of FLOT neoadjuvant treatment and had to stop neoadjuvant treatment and undergo surgery due to gastric bleeding. One patient suffered from acute myocardial infarction after the first cycle of treatment with FLOT and after that he discontinued neoadjuvant treatment. 2 patients received only 3 cycles of neoadjuvant treatment and went on to receive surgery due to worsening clinical conditions. One patient suffered from febrile neutropenia after first cycle of FLOT neoadjuvant treatment and continued for the remaining 3 cycles of treatment with FOLFOX instead. Our results are quite comparable to FLOT4-trial data where 91% of the treated population was able to complete the intended neoadjuvant treatment.

These results were added to the text.

3- In the results the authors state:"Out of 66 patients who received surgery after perioperative chemotherapy, only 57 160 (86%) received adjuvant chemotherapy after pre-operative management. "

Can the authors add some details about the reasons why patients did not received adjuvant FLOT?

REPLY: 9 patients were not able to receive adjuvant chemotherapy with FLOT. 3 patients were lost at follow-up after surgery and did not receive any post-operative treatment. 5 patients had such poor clinical conditions due to weight loss and loss of muscle mass after surgery that they were deemed not able to receive any post-operative treatment. One patient died after surgery due to reasons not directly associated to the disease or treatment (traumatic injury).

These results were added to the text.

4- Please provide a table about univariate and multivariate analysis results.It will be easier for the readers instead of going through all the text.

REPLY: We have provided another table (table 2) that summarises the results of both univariate and multivariate analysis.

5- In the discussion section the authors pointed out that: " Interestingly, cN stage had only a minor impact on survival outcomes." How can the authors explain this contradictory result comparing it with the available evidences?

REPLY: Indeed, we were also puzzled by this result. However, the number of patients who were cN0 were relatively small compared to the whole group of patients treated with perioperative FLOT (15/76, 20%). There was a trend towards better overall survival in patients with cN0 status but did not reach statistical significance. We believe that the lack in impact on overall survival might be more due to the relatively small sample size of patients with negative lymph nodal involvement more than anything else. However, it must also be said that all evidences that we currently have in patients with GEJ concerning comparison between FLOT and chemoradiotherapy (p.e. Neo-AEGIS trial) have shown that FLOT chemotherapy compared to other options is less effective on achieving downstaging of lymph node involvement: because of this fact we can’t rule out that lymph node involvement might not be as relevant, as opposed to other clinical factors, on the prognosis of patients treated with FLOT perioperative chemotherapy.

6- Concerning the surgical radically the authors comment in the discussion : "Quality of surgical assessment before treatment start is therefore crucial, as it defines the likelihood of radicality: in our  group of patients that were initially deemed able to undergo radical surgery, only 66/76 302 (85%) were ultimately resected, thus suggesting that there is dire need of improvement in this aspect. "

How can the authors explain a surgical radically of 71% with a ypT4a of 22% and a ypT4b of only 6%? Did they experience a R1 or R2 resection of all T4 patients?

This data could be a tremendous bias about survival analysis. Moreover I could not find, even in the supplementary materials, any data about radically od surgery in the historical cohort of patients. Which was the R0 resection rate in the adjuvant CHT group?

REPLY: Thank you for pointing out at this crucial point: in the adjuvant chemotherapy group only 1 patient received R2 resection whereas out of the remaining 94 patients, 76 (81%) received R0 resection. The remaining 16 (17%) patients received R1 resection. There was a statistically significant impact on OS (mOS respectively 76 vs 53.74 vs 11.80 months for R0 vs R1 vs R2, p<0.0001). This was added in discussion section.

On the other hand, in the FLOT perioperative cohort, 3 (4%) patients received R2 resection, 9 (12%) patients received R1 resection and the remaining 54 (71%) received R0 resection (and additional 10 patients did not receive any resection due to occurrence of distant metastases or patient’s worsening clinical conditions. 2/3 (66%) patients who received R2 resection had cT4 stage at diagnosis whereas 6/9 (66%) patients who received R1 resection had cT4 stage at diagnosis. Albeit there is a trend towards increased risk of R1/R2 resection in patients with cT4 stage at diagnosis, the difference was not statistically significant at chi-square test (p=0.451).

The difference in likelihood to receive R0/R1/R2 resection was not significantly different between the group of patients who received FLOT neoadjuvant treatment followed by surgery followed by FLOT adjuvant treatment vs patients who received surgery followed by adjuvant treatment (p=0.3078).

If we consider the whole population of patients, as to include also patients who received neoadjuvant FLOT and who did not receive any surgery, there is a statistically significant difference in outcome compared with patients who received surgery followed by adjuvant treatment (p=0.0017).

This difference in outcome is obviously related to the number of patients who were deemed initially as resectable and that after the end of neoadjuvant treatment were deemed not resectable. Lack of proper staging at the beginning of neoadjuvant treatment (by means of chest-abdomen CT scan and echoendoscopy in patients deemed eligible for surgery) might have overlooked metastatic spread in the peritoneum that could have been assessed properly only by means of laparoscopy (that was not performed routinely in all patients) and that might have excluded patients from surgical resection of the primary tumor after the end of neoadjuvant treatment. In addition to that, too much optimistic attitude concerning management of patients with locally advanced disease and the likelihood to response to neoadjuvant FLOT chemotherapy might have led surgeons to deem as initially resectable patients who did have such disease extent that would be otherwise considered not amenable to surgery.

We acknowledge this limitation of our analysis and we added this considerations to the discussion section. It must be said that the only way to reduce this bias would be to include in the historical cohort of patients also those cases who were initially deemed as resectable and that were not ultimately resected upon surgery and who received palliative chemotherapy (and it would be extremely difficult to identify these patients) or assess the issue of perioperative FLOT vs surgery followed by adjuvant chemotherapy in a prospective fashion (and it would require a rather considerable follow-up time in order to see the differences between the two groups of patients).

7- Please pay attention about the Privacy Policy of this journal. In the Supplementary 2 Adjuvant cohort database there are sensitive data (name of the patients, date of birth, etc...)

REPLY: Thank you for having noticed our mistake. We removed sensitive data from the database in order to comply to Privacy Policy of the journal.

Reviewer 2 Report

The study aimed to assess the impact of the presence of signet ring cells on overall survival and disease free survival in patients who received either perioperative or adjuvant chemotherapy. The main finding was that perioperative chemotherapy was associated with shorter survival than adjuvant chemotherapy in patients with presence of signet ring cells. This is clinically important as it suggests that these patients may benefit from surgery without any delay.

I have the following major remarks:

1)     The study had several limitations, including a small sample size (n=76). In the presentation of the results with KM curves with numbers at risk, none of the these add up to 76 at start (0 months).

2)     The FLOT cohort (2017-2022) and the historical cohort (2005-2020) overlapped considerably. This seems problematic considering the retrospective study design and increase the risk of selection bias?

3)     What was the number of patients lost to follow-up?

4)     Histological classification is unclear and lacks description in the methodology section. If the main point is to assess the significance of signet ring cells rather than Lauren subtypes this should be stated in the introduction with a few references to similar studies.

Signet ring morphology (WHO) has previously been considered to correspond with and to be a subclass of Lauren diffuse type cancers. In this study, the number of patients with signet ring cells was identical to the number of patients with diffuse type cancer. It is likely that patients from Lauren classes other than diffuse type were also found to contain signet ring cells.

The number or proportion of signet ring cells in a cancer before it was considered “signet ring” should be described.   

In table 1 “undifferentiated” is listed as a subtype of Lauren class and constituted 30% of the cases. “Undifferentiated” is usually not a subtype of Lauren, but “indeterminate” is.

The manuscript would benefit from careful proof-reading, language and syntax editing.

Some examples include:

·        Please change «not-metastatic» to non-metastatic throughout the entire manuscript.

·        “Worse OS”: please change to shorter OS.

·        “mOS:20.36vs46.92 months,HR:3.30,95%CI:1.56-6.99,p=0.0018» Please use spaces in the presentation of results, delete the abbreviation mOS, limit to one decima in the number of months.

·        Line 63: “In Western countries instead”. Please delete or replace “instead”.

·        Linge 69: “benefit less” should be replaced by “do not benefit”, since also a minor benefit would be worthwhile.

·        Line 79: what are “gastric/GEG cancer”? GEG has not been defined. Do the authors med gastroesophageal junction (GEJ) cancers?

·        Line 149: the percentages of Lauren histological class do not add up to 100% (30%+34%+5%), please explain why not.

·        Line 152: what is “scarcely differentiated”? Do the authors mean poorly differentiated?

·        Line 95: please explain why cT2 N+ patients were not included in analyses.

·        Line 98: please define ECOG

·        Please avoid the word “confirm” when presenting results from the multivariable analysis in a small retrospective study.  None of the analyses done prove causality, the more careful “supported by” suits the study design better.

Line 149 and onwards: The number of patients (n) and % do not match. 24 of 76 patients is 31.57% which should be 32% as integer, not 30% as stated in the text (line 149 and Table 1), not 31% as stated in line 150, which leaves an unreliable impression.

Please see general comments

Author Response

Thank you for your constructive criticism. Please find below the replies to your comments:

The study aimed to assess the impact of the presence of signet ring cells on overall survival and disease free survival in patients who received either perioperative or adjuvant chemotherapy. The main finding was that perioperative chemotherapy was associated with shorter survival than adjuvant chemotherapy in patients with presence of signet ring cells. This is clinically important as it suggests that these patients may benefit from surgery without any delay.

I have the following major remarks:

  • The study had several limitations, including a small sample size (n=76). In the presentation of the results with KM curves with numbers at risk, none of the these add up to 76 at start (0 months).

REPLY: We acknowledge that the small sample size is the most relevant limitation of our analysis. The reason that explains KM curves with different numbers of risk is related to missing data concerning survival estimates for that specific factor or the impossibility to define in a dichotomous way one factor (p.e. concerning histology how to define the subset of patients who have mixed histology). We would also like to clarify that the incidence of gastric cancer in Italy was estimated in 2020 to be around 14000 new cases/year (with decreasing trend of incidence); assuming that gastric cancer incidence is evenly distributed in our Country and considering the number if inhabitants of the whole Marche region where this study was conducted, expected incidence of gastric cancer is about 300-350 new diagnoses/year. Considering that screening for gastric cancer is not performed, leading to a considerable number of patients who are diagnosed with either metastatic or locally advanced stage of disease involvement leading up to palliative chemotherapy and that would not be candidate to receive perioperative management, we believe that this study fully encompasses the whole state of the art of perioperative treatment for patients with not-metastatic disease in our geographic region.  

  • The FLOT cohort (2017-2022) and the historical cohort (2005-2020) overlapped considerably. This seems problematic considering the retrospective study design and increase the risk of selection bias?

REPLY: We find your comment quite accurate: indeed, the overlap between the two groups of patients is quite evident. However, if we consider that we are assessing relative differences in OS between these two groups of patients we should also take into account that for all patients that experience disease relapse palliative chemotherapy might be further prescribed and might have also an impact on OS. Since treatment for metastatic gastric cancer patients has changed considerably in the last decade, if we would have compared a historical cohort of patients that was entirely made up of patients that did not overlap with FLOT cohort we would have had a HIGHER risk of selection bias as we would have included patients that would inherently have less survival times upon relapse due to having less treatment option at their disposal. Indeed, since FLOT introduction there has been a trend towards increased use of perioperative chemotherapy vs surgery followed by adjuvant chemotherapy compared to times prior FLOT4 publication; however, several guidelines during that time period suggested that either approach could be used (as proper comparison between the 2 treatment strategies is lacking).

  • What was the number of patients lost to follow-up?

REPLY: 3 patients were lost at follow-up. All 3 patients were lost at follow-up after that they received surgery and did not receive any adjuvant chemotherapy. All other patients are currently in active follow-up (for the sake of the calculations presented in this study data cut-off has been set at October 2022). This has been added to the text.

4)     Histological classification is unclear and lacks description in the methodology section. If the main point is to assess the significance of signet ring cells rather than Lauren subtypes this should be stated in the introduction with a few references to similar studies.

Signet ring morphology (WHO) has previously been considered to correspond with and to be a subclass of Lauren diffuse type cancers. In this study, the number of patients with signet ring cells was identical to the number of patients with diffuse type cancer. It is likely that patients from Lauren classes other than diffuse type were also found to contain signet ring cells.

The number or proportion of signet ring cells in a cancer before it was considered “signet ring” should be described.  

In table 1 “undifferentiated” is listed as a subtype of Lauren class and constituted 30% of the cases. “Undifferentiated” is usually not a subtype of Lauren, but “indeterminate” is.

REPLY: Thank you for your comment. We have changed the methods section in order to reduce confusion: as a matter of fact, as you have already stated, we have considered as “signet-ring” morphology all those instances where pathologists described signet-ring cells presence in more than 50% of the tumor.  All patients that had properly defined diffuse-type gastric cancer were included in the definition of “diffuse-type” gastric subtype, regardless of the presence/absence of signet-ring cells. The apparent mismatch of numbers is related to the fact that in 2/4 mixed subtype gastric cancer specimens the presence of signet-ring cells was deemed to be particularly relevant (75%) as to consider these 2 cases into the signet-ring cell positive group (and obviously not included in the properly defined diffuse type group). In addition to that, 3 patients with undifferentiated carcinoma did display a relevant number of signet-ring cells that led us to include them in the “signet-ring cell” subgroup. Finally, as strange as it might sound, one case that displayed histological features that led up to diagnosis of “intestinal type” gastric cancer displayed also a relevant number of signet-ring cells that led us to include this case into the “signet-ring cell” subgroup. In table 1, rather than simply showing the group of patients who did have “indeterminate” histology by Lauren classification, we decided to specify which subtype by WHO classification they belonged to (majority was undifferentiated by WHO classification, with a minor number of patients having either mucinous or squamous type gastric cancer). Following your suggestion we decided to just define them as “indeterminate” histology, as proper by Lauren classification.

The manuscript would benefit from careful proof-reading, language and syntax editing.

Some examples include:

  • Please change «not-metastatic» to non-metastatic throughout the entire manuscript.

REPLY: Thank you for pointing that out. We changed that.

  • “Worse OS”: please change to shorter OS.

REPLY: We have changed that.

  • “mOS:20.36vs46.92 months,HR:3.30,95%CI:1.56-6.99,p=0.0018» Please use spaces in the presentation of results, delete the abbreviation mOS, limit to one decima in the number of months.

REPLY: We have used spaces as suggested and limited to one decima the number of months. However, we decided to leave mOS (and it was changed into MEDIAN OS) as to specify what we are referring to.

  • Line 63: “In Western countries instead”. Please delete or replace “instead”.

REPLY: We deleted “instead”.

  • Linge 69: “benefit less” should be replaced by “do not benefit”, since also a minor benefit would be worthwhile.

REPLY: We agree to that. We changed that into “do not benefit”

  • Line 79: what are “gastric/GEG cancer”? GEG has not been defined. Do the authors med gastroesophageal junction (GEJ) cancers?

REPLY: Exactly. That was a typo. It was corrected.

  • Line 149: the percentages of Lauren histological class do not add up to 100% (30%+34%+5%), please explain why not.

REPLY: Percentages do not add up to 100% because there was a typo in the text. This has been corrected.

  • Line 152: what is “scarcely differentiated”? Do the authors mean poorly differentiated?

REPLY: Undifferentiated (by WHO classification). This has been corrected.

  • Line 95: please explain why cT2 N+ patients were not included in analyses.

REPLY: I don’t understand your comment. Patients with cT2 cN+ WERE included in the analysis (as well as one patient with cT1 cN+ tumor for that matter). Inclusion criteria properly state that we included patients who did have cT3 or higher T stage AND/OR cN+ disease. That means that ANY patient with cN+ disease was included.

  • Line 98: please define ECOG

REPLY: We defined ECOG.

  • Please avoid the word “confirm” when presenting results from the multivariable analysis in a small retrospective study. None of the analyses done prove causality, the more careful “supported by” suits the study design better.

REPLY: I do not entirely agree with your statement: multivariate analysis is not used to prove causality by any means, even when performed in “larger” retrospective studies. We used the word “confirm” as to convey the fact that out of a series of variables that were found to be associated with a statistically significant impact on survival outcomes, there were a few of them that seemingly maintained their statistically significant impact on survival outcomes after multivariate testing and others that did lose such impact. We changed the way it is written as to suit your wishes.

Line 149 and onwards: The number of patients (n) and % do not match. 24 of 76 patients is 31.57% which should be 32% as integer, not 30% as stated in the text (line 149 and Table 1), not 31% as stated in line 150, which leaves an unreliable impression.

REPLY: Thank you for pointing that out. There was a typo and now it has been corrected.

Round 2

Reviewer 1 Report

In the authors' reply to comment n. 6, they wrote:" If we consider the whole population of patients, as to include also patients who received neoadjuvant FLOT and who did not receive any surgery, there is a statistically significant difference in outcome compared with patients who received surgery followed by adjuvant treatment (p=0.0017)."

Which outcome are they referring to: survival or R0 resection rate? Please try to make it clearer in the text.

Author Response

I would like to express our thanks on behalf of all co-authors to both reviewers as I believe that they have helped to improve our manuscript. Please find below the replies to the second round of request for revisions:

Reviewer 1:

In the authors' reply to comment n. 6, they wrote:" If we consider the whole population of patients, as to include also patients who received neoadjuvant FLOT and who did not receive any surgery, there is a statistically significant difference in outcome compared with patients who received surgery followed by adjuvant treatment (p=0.0017)."Which outcome are they referring to: survival or R0 resection rate? Please try to make it clearer in the text.

REPLY: You make a very good point. We were referring to overall survival. This has been changed in the text as to make it clearer.

Reviewer 2 Report

I thank the authors for revising the manuscript. 

The inserted Table 2 has large font size and is difficult to read for reviewers as well as others. Please correct. 

Please be consistent using only one decimal when presenting OS in months (line 45). 

-

Author Response

I would like to express our thanks on behalf of all co-authors to both reviewers as I believe that they have helped to improve our manuscript. Please find below the replies to the second round of request for revisions:

Reviewer 2:

 I thank the authors for revising the manuscript.

The inserted Table 2 has large font size and is difficult to read for reviewers as well as others. Please correct.

REPLY: We have acknowledged that. Table 2 was shrinked down and font was also changed in order to make it easier to browse the results.

Please be consistent using only one decimal when presenting OS in months (line 45).

REPLY: Thank you for having pointed at that out. We have corrected it.